# The Potential Role of Plant Polysaccharides in Treatment of Ulcerative Colitis

**DOI:** 10.3390/pharmaceutics16081073

**Published:** 2024-08-16

**Authors:** Yilizilan Dilixiati, Adila Aipire, Ming Song, Dilaram Nijat, Abudukahaer Wubuli, Qi Cao, Jinyao Li

**Affiliations:** Xinjiang Key Laboratory of Biological Resources and Genetic Engineering, College of Life Science and Technology, Xinjiang University, Urumqi 830017, China; 107552100963@stu.xju.edu.cn (Y.D.); adila16@xju.edu.cn (A.A.); 107552100985@stu.xju.edu.cn (M.S.); dilaram@xju.edu.cn (D.N.); 107552103429@stu.xju.edu.cn (A.W.); 107552303689@stu.xju.edu.cn (Q.C.)

**Keywords:** plant polysaccharides, inflammatory bowel disease, ulcerative colitis, anti-inflammatory

## Abstract

Ulcerative colitis (UC) results in inflammation and ulceration of the colon and the rectum’s inner lining. The application of herbal therapy in UC is increasing worldwide. As natural macromolecular compounds, polysaccharides have a significant role in the treatment of UC due to advantages of better biodegradation, good biocompatibility, immunomodulatory activity, and low reactogenicity. Therefore, polysaccharide drug formulation is becoming a potential candidate for UC treatment. In this review, we summarize the etiology and pathogenesis of UC and the therapeutic effects of polysaccharides on UC, such as regulating the expression of cytokines and tight junction proteins and modulating the balance of immune cells and intestinal microbiota. Polysaccharides can also serve as drug delivery carriers to enhance drug targeting and reduce side effects. This review provides a theoretical basis for applying natural plant polysaccharides in the prevention and treatment of UC.

## 1. Introduction

Inflammatory bowel disease (IBD) is a chronic condition involving persistent inflammation of the gastrointestinal tract. It includes two primary forms: ulcerative colitis (UC) and Crohn’s disease (CD). UC was traditionally considered a disease predominantly found in Western countries. However, there has been a rise in the incidence of UC in other developing countries, for example, Asian countries. UC impairs quality of life, and patients may have proximal disease extension. Over the past decade, the treatment options for UC have greatly expanded with the introduction of biologics and small molecules [1], while clinical and endoscopic remissions of these treatments are also employed. Despite these advances, remission rates do not surpass 20–30% in induction clinical trials [1]. At present, the drugs for UC mainly include 5-amino salicylic acids, glucocorticoids, immunosuppressants, micro-ecologics, biological agents, such as tumor necrosis factor (TNF) antagonists, anti-integrin agents, and Janus kinase (JAK) inhibitors [2]. However, these drugs used to treat UC have limited efficacy or induce severe adverse reactions [3]. Therefore, there is an urgent need to discover new therapies that are both curable and tolerable for patients with UC. Based on the shortcomings of conventional drugs, research and development of herbal medicine has been initiated in the field of UC treatment. Herbal medicines, including *Aloe vera* [4], *Triticum aestivum* [5], *Andrographis paniculate* [6], and *Boswellia serrata* [7] are known for their therapeutic effects, as well as their safety and biocompatibility.

Many immunomodulators based on herbal plant extracts can enhance the defense response of the animal organism, which is an effective way for the organism to increase its resistance to various diseases [8]. Plant polysaccharides, as natural functional biomolecules, are showing a growing trend in the treatment of UC due to their unique physicochemical properties and biological activities [9]. Polysaccharides are complex macromolecules composed of more than ten monosaccharides linked by glycosidic bonds, forming linear or branched chains with high molecular weight [10,11]. Depending on whether they are absorbed in the intestine, polysaccharides can exert their biological effects through both microbiota-dependent and -independent mechanisms [10,12]. Some polysaccharides are easily digested and absorbed by the intestines, allowing them to reach the immune organs through blood circulation where they exert immune regulatory functions. A proportion of polysaccharides are also resistant to digestion and are mainly metabolized by the gut microbiota in the large intestine, modulating the gut microbiota diversity and compositional structure of the gut microbiota. In this review, to provide a reliable basis for polysaccharides in the treatment of UC and the development of functional foods or drug candidates, we summarized the pathogenesis of UC and the therapeutic role of plant polysaccharides and their mechanism in UC.

## 2. Etiology and Pathogenesis of UC

Inflammation is a typical defense mechanism against response to infection, tissue damage, or harmful stimuli, which is a complex process triggered by various factors [13]. Inflammatory diseases, as a group of clinical disorders, are marked by abnormal inflammatory responses, with chronic inflammation being a significant hallmark [14]. 

As a chronic inflammatory condition, UC leads to inflammation and ulceration of the colon and the rectum’s inner lining [15]. UC typically manifests as diarrhea, abdominal discomfort, fatigue, decreased appetite, weight loss, and anemia [15]. The etiology and pathogenesis of UC are driven by gene–environment interactions, which involve genetic predisposition, defects in the epithelial barrier, dysregulated immune responses, microbial dysbiosis, and environmental factors [11] (Figure 1). A positive family history has been reported in 1.5% to 24% of cases [16]. Although numerous studies on genetic risks in UC have been performed, an accurate genetic risk assessment still needs to be conducted. At present, there are more than 200 gene variants related to inflammatory bowel disease [11]. In UC, studies have identified the involvement of the human leukocyte antigen (HLA) complex [17], including *HLA-DRB1*0103* [18] and *HLA-DRB1*1502* [19], conferring the most significant risk. Certainly, an altered immune response is critical in determining the pathogenesis of UC. The main features of gut microbiota (GM) and gut-associated lymphoid tissue (GALT) are shifting towards an inflammatory pattern. For example, T regulatory (Tregs) cells in GALT are reduced with an increase in T helper cell 17 (Th17) and T helper 1 (Th1) cells and enhancement of TNF and interleukin-1β (IL-1β) production. Additionally, the innate immune system is activated through the triggering of toll-like receptor (TLR)-4, while the epithelium is directly damaged, further perpetuating inflammation [1,20].

The intestinal mucosal barrier is primarily formed by the tight junctions (TJs), which form complexes between neighboring intestinal epithelial cells (IECs). Modulation of TJs is a potent strategy for increasing absorption, but inflammation often causes disruption of the TJ barrier [21]. The TJs are related to several transmembrane and cytosolic proteins, including occludin, claudins, zonula occludens (ZOs), tricellulin, cingulin, and junctional adhesion molecules (JAM) [22]. The increasing number of studies on colitis models have identified the important role of TJ proteins in the pathogenesis of UC [23].

Mucus, synthesized and secreted by goblet cells, is a critical component of intestinal barrier integrity, as the mucus covers the surface of the intestinal lining [24]. This gel-like substance acts as a barrier between the contents of the intestine and the host tissue, providing a habitat for the gut microbiota and safeguarding the intestinal lining [25]. Studies have shown that the expression of mucin, a key component of mucus, is elevated in patients with UC [26]. Meanwhile, the bacteria and their structures can influence the properties of the mucus barrier in some ways, impacting both health and disease outcomes [27,28,29,30]. A healthy balance of microbiome community is vital for maintaining mucus barrier homeostasis, which involves a dynamic balance of production, secretion, expansion, and proteolysis of mucus components [30]. Yuichiro Nishihara et al. [28] used 16S rRNA-based analysis of rectal mucosal tissues to determine the relationship between the composition of mucosa-associated GM and the clinical progression. The less microbial diversity during the active period of UC is mainly characterized by decreased Bacteroidetes [29], *Eubacterium rectale*, and *Akkermansia muciniphila*, and increased *E coli* [30]. 

Environmental factors and dietary habits may predispose an individual to UC by influencing the diversity and composition of the bacteria, viruses, and fungi that form the intestinal microbiome [31]. Western relevant studies have shown that current smoking status is associated with a reduced risk of developing UC [32]. Orholm et al. [33] reported that differences in smoking patterns among patients seemed to be of significance for the occurrence of discrepancies in UC. Nevertheless, the effect of smoking on UC in Asian populations remains uncertain [34]. 

In addition, the gastrointestinal tract plays an important role in the human body by providing a nutrient and breeding environment for GM, which in turn provides nutritional, metabolic, and immune benefits [35]. It is undeniable that diet likely plays a significant role in the development and progression of UC. High intakes of mono-unsaturated fats, polyunsaturated fats, and vitamin B6 may increase the risk of developing UC [36]. Western-type diets rich in saturated fats, in particular, may contribute to a higher prevalence of complex immune-mediated diseases like IBD in genetically susceptible hosts [37]. Aside from dietary factors, personal habits have been linked to the development of UC. Ashwin N et al. [38] found that less than 6 h sleep/day and more than 9 h sleep/day are each linked to a higher risk of UC, according to the data from the Nurses’ Health Study (NHS) I and II. However, sleep disturbance was not associated with an increased risk of UC [38]. The connection between sleep and UC needs to be further verified. 

## 3. The Role of Natural Polysaccharides in UC

In recent years, natural polysaccharides for protecting intestinal health have become a topic of intestine research. Therefore, researchers have devoted huge amounts of effort to investigate the role of natural plant-resourced polysaccharides in UC. As shown in Figure 2, polysaccharides can intervened in UC through various ways, such as the regulation of inflammatory cytokines, intestinal flora, immune cell balance [39]. Zhang Y et al. [40] found that polysaccharides could be absorbed through the small intestine followed by accumulation in the liver and kidney. Nevertheless, non-digestible polysaccharides can reach the colon, where they can be fermented by the gut microbiota, acting as a prebiotic [41]. As shown in Table 1, polysaccharides from different herbal plant sources could ameliorate UC differently. The detailed mechanisms of plant-derived polysaccharides in UC are summarized as follows.

### 3.1. Regulating the Expression of Cytokines

Cytokines are synthesized and secreted by stimulated immune cells (e.g., monocyte, macrophage, T cells, B cells, NK cells, etc.) or some non-immunocytes (e.g., endothelial cells, skin cells, fibroblasts) [53]. Potential strategies for treating inflammation include inhibiting the release of inflammatory mediators, correcting the dysregulation of proinflammatory cytokines like IL-1β, TNF-α, and IL-6, and upregulating anti-inflammatory cytokines such as IL-10. Meanwhile, in the intestines of patients with UC, the overexpression of proinflammatory cytokines leads to ongoing mucosal inflammation and plays a direct role in the development of UC [61]. Research has shown that many polysaccharides inhibit inflammation-related cytokines and nuclear factor-kappa B (NF-κB) signaling pathways both in vitro and in vivo, causing potent anti-inflammatory effects [62,63,64]. As known as a major inflammatory pathway, NF-κB is closely related to most kinds of inflammatory disease [65].

As a key mediator in the inflammatory response, TNF-α is a pleiotropic proinflammatory cytokine involved in various cellular processes, such as cell proliferation, survival, and death [66]. However, Aloe polysaccharide can effectively reduce the lipopolysaccharides (LPS) and TNF-α induced HT-29 cell apoptosis and secretion of IL-6; its mechanisms may involve the down-regulation of phosphorylated Janus kinase 2 (p-JAK2) and phosphorylated signal transducer and activator of transcription 3 (p-STAT3) expression [47]. In addition, *Schisandra chinensis* polysaccharide (SCP) increased the IFN-γ and IL-4 levels and decreased IL-6, IL-17, IL-23, and TNF-α, significantly [49]. In conclusion, polysaccharides extracted from plants could regulate the balance of proinflammatory and anti-inflammatory cytokines, which is an important way of alleviating UC.

### 3.2. Regulating the Balance of Immune Cells

The abnormal accumulation of cytokines and chemokines from the immune cells or non-immunocytes in the intestinal mucosa can trigger inflammatory responses. Various cells, including antigen-presenting cells (dendritic cells and macrophages), effector T cells, and regulatory T cells, play crucial roles in UC pathogenesis by either promoting or inhibiting inflammation [11]. Naïve CD4+ T cells differentiate into effector CD4+ T cells, such as Th17 cells and Tregs, through the secretion of various cytokines [67]. Typical effector Th response and Tregs imbalance result in the dysregulated production of several key cytokines, including TNF(Th1), IL-5, IL-6, IL-13(Th2), IL-17, IL-21, IL-22(Th17) [1]. Specifically, Th17 cells infiltrate the gastrointestinal mucosa and produce excessive inflammatory cytokines, such as IL-17A, initiating a more intense inflammatory response that Tregs are not able to tolerate in UC patients.

In another study, *Pseudostellariae* polysaccharide significantly decreased the infiltration of lymphocytes and neutrophils, resulting in attenuating the pathological damage in colitis [58]. In addition, polysaccharides extracted from *Atractylodes macrocephala* (PAMK) regulated the balance between Th 17 and Tregs in the mesenteric lymph nodes (MLN) and spleen in mice with colitis, which is a way that is dependent on the inhibition of the IL-6/STAT3 signaling pathway [59]. Poria cocos polysaccharide has shown significant therapeutic effects on UC by inhibiting the IL-33/ST2 signaling pathway and reducing mesothelial cell (MC) activation, thus inhibiting the expression of inflammatory factors and reducing the degree of colonic inflammatory infiltration [68].

### 3.3. Regulating the Function of Intestinal Mucus

The intestinal mucosal barrier, primarily formed by the tight junctions (TJs) of IECs, serves as the first defense against a hostile environment. There is a strong relationship between intestinal microflora and the intestinal mucosal barrier. Intestinal microflora dysbiosis weakens the mucosal barrier, enhances intestinal bacterial translocation, and disrupts structural barriers by altering intestinal TJ proteins [69]. Relevant studies showed that TJs could prevent the spread of pathogens and harmful antigens across the epithelium [70]. Upregulated expressions of these proteins could alleviate the damage to the intestinal barrier. Noni fruit polysaccharide (NFP) promotes mucosal and tight junction protein (ZO-1 and occludin) expression, thereby reducing the damage to the colonic mucosal barrier caused by dextran sulfate sodium salt (DSS) [50]. In the same way, polysaccharides from *Scutellaria baicalensis* [43], *Gracilaria lemaneiformis* [55], and *Angelica sinensis* [57] also could promote the production of ZO-1 and occludin-1. In conclusion, polysaccharides appear to protect the integrity of the TJ network by regulating the TJ protein.

The villus–crypt axis is considered the fundamental functional unit of the intestinal mucosa. The ratio of villus height to crypt depth (V/C) is also regarded as a key indicator of intestinal morphology [71]. According to Chen et al research, treating with cyclophosphamide (CTX) exhibited severe damage to the intestinal villi and crypt structures. However, this damage was found to be alleviated in a dose-dependent manner by *Millettia Speciosa* polysaccharides (MSCP) [72]. This was comparable to the improvement in goblet cells and mucin observed with the synergistic ginseng polysaccharides and ginsenosides [35]. 

Goblet cells are a type of specialized secretory intestinal epithelial cell (IECs) that secretes a variety of mucins. According to expressed tissues and cell types, they are broadly categorized into two types: secretory and membrane-related mucins. Mucin (MUC)-2, MUC-5AC, MUC-5B, and MUC-6 are gel-forming secretory mucins found on the mucosal surface, while MUC-1, MUC-3, MUC-4, MUC-13, and MUC-17 are membrane-associated mucins found in the apical membranes of epithelial cells [73]. A decreased mucin expression is observed in UC patients compared to healthy individuals [26]. Defects in the expression of proteins in the colon, such as MUC1 and MUC2, have been linked to a higher susceptibility to chronic inflammation, highlighting the importance of mucins in repairing the intestinal barrier [74]. Several studies have demonstrated that MUC2 enhances gut homoeostasis and oral tolerance. It has been observed that MUC2 influences dendritic cells (DCs) and intestinal epithelial cells. Additionally, the MUC2 receptor complex has been found to inhibit inflammatory responses in DCs [75]. MSCP was found to enhance the expression of genes related to intestinal mucosal integrity, such as Occludin1, Claudin1, and MUC2, thereby attenuating CTX-induced intestinal barrier damage in mice [72].

### 3.4. Protecting the Intestinal Barrier Integrity

The intestinal epithelium is equipped with a defense barrier comprising tightly connected IECs, which serve to prevent the invasion of pathogenic microorganisms [72]. Aloe polysaccharide reduced the apoptosis of IECs, possibly through the inhibition of the JAK2/STAT3 signaling pathway [47], which means that the polysaccharide could alleviate the negative effect of UC. In the gastrointestinal tract, apoptosis is typically limited to superficial epithelial cells. However, apoptosis can be more extensive in pathological states of inflammation or infection [52]. In another study, Rheum tanguticum polysaccharide (RTP) significantly decreased malondialdehyde (MDA) levels, lactate dehydrogenase (LDH) activity, and cell apoptosis in IECs, suggesting that RTP may protect against H_2_O_2_-induced IEC injury [52]. Polysaccharides from *Morinda citrifolia* also had a similar function to restore GSH colonic level and decreased MDA concentration in colitis induced by acetic acid [76]. 

NOD-like receptor 3 (NLRP3) is a member of the NOD-like receptor (NLR) family and would be expressed in immune cells and IECs, which could be activated by numerous complexes such as LPS and ATP. Therefore, the regulation of NLRP3 inflammasome and β-arrestin1 signaling pathways may be potential therapeutic targets for UC treatment. The relevant studies discussed the role of polysaccharides from Dendrobium officinale (DOPS) in the treatment for UC, and the results showed that DOPS could be related to the inhibition of NLRP3 and β-arrestin1 signaling pathways [46]. Similarly, crude polysaccharides (QHPS) extracted from a two-herb formula consisting of Lycium barbarum and Astragalus membranaceus, helped maintain intestinal mucosal integrity and repaired intestinal tract injuries by lowering serum levels of endotoxin (EDT), diamine oxidase (DAO), and D-lactate (DLA) in the serum [48]. In conclusion, specific polysaccharides from natural plants can alleviate UC symptoms by restoring intestinal mucosal barrier functions.

### 3.5. Regulating the Intestinal Microbiota

Intestinal microbiota constitute a complex community that interact with each other and with the host to modulate biological processes essential for health. Changes in the metabolic products of gut microbiota are considered to modulate the integrity of the epithelial layer and the immune response of the gut [77]. Research indicates that harmful interactions between the intestinal immune system and gut microbes can worsen UC [78]. Gut microbes, mainly Bacteroidetes and Firmicutes, play a crucial role in human health, including regulating host immunity, maintaining intestinal barrier integrity, and inhibiting intestinal pathogens [79]. The Bacteroidetes phylum is comprised of three genera: *Bacteroides*, *Prevotella*, and *Xylanibacter*. The phylum Firmicutes is characterized by the presence of numerous genera, including *Lactobacillus*, *Ruminococcus*, *Clostridium*, *Eubacterium*, *Fecalibacterium*, and *Roseburia*; *Escherichia* and *Desulfovibrio* belong to the Proteobacteria, while the *Akkermansia* genus is classified within the *Verrucomicrobia phylum* [59]. 

Gut microbiota can drive pathogenicity via two types of mechanisms: expansion of ‘pro-inflammatory’ species or restriction in the protective compounds of the microbiota. In mucosal biopsies, some bacteria were more frequently found in UC patients (*Rhodococcus*, *Shigella*, *Escherichia coli*, and *Stenotrophomonas*) than in siblings unaffected by UC [80]. Bacteroidetes, at both the genus and phylum levels, significantly elevated the levels of acetic acid, propionic acid, butyric acid, and total short-chain fatty acids (SCFAs) [49]. Most of the consumed soluble dietary fiber is spontaneously fermented by colonic bacteria to produce organic acids, including SCFA, which is a crucial energy source of colonocytes [56]. SCFAs, produced by microbes, can affect intestinal health via multiple targets, including cell metabolism, proliferation, and immune response [79]. The prominent SCFAs-producing genera *Akkermansia* and *Blautiawere* increased after subacute ulcerative colitis (SUC) mice metabolizing Aloe polysaccharides (APs), and *Bacteroidetes* was significantly related to acetic acid, propionic acid, inflammatory cytokines by Spearman analysis [79]. Polysaccharide extracted from *Arctium lappa* (ALP-1) could increase the abundance of *Firmicutes, Ruminococcaceae, Lachnospiraceae*, and *Lactobacillus*. Meanwhile, ALP-1 significantly reduced the levels of Proteobacteria, Alcaligenaceae, Staphylococcus, and Bacteroidetes. The abundance of *Firmicutes*, *Bifidobacterium*, *Lactobacillus*, and *Roseburia* was also increased significantly by *Scutellaria baicalensis* polysaccharide. At the same time, this polysaccharide could significantly inhibit the levels of *Bacteroides*, *Proteobacteria*, and *Staphylococcus* [43]. In Zhou’s work, a high dose of *Lonicera japonica* polysaccharide (LJP) had significant effects on improving the intestinal probiotics (*Bifidobacterium* and *Lactobacilli*) and antagonizing the pathogenic bacteria (*Escherichia coli* and *Enterococcus*) [81]. Based on Jin-Hua Tao’s research, after the treatment with polysaccharides from *Chrysanthemum morifolium*, the abundance of opportunistic pathogens (*Escherichia*, *Enterococcus* and *Prevotella*) in UC rats was decreased. At the same time, the levels of protective bacteria, such as *Butyricicoccus*, *Clostridium*, *Lactobacillus* and *Bifidobacterium*, *Lachnospiraceae* and *Rikenellaceae*, were elevated to various degrees [54]. In another work, after treating UC rats with *Radix Pseudostellariae* polysaccharide, the content of SCFA in UC rats was increased. Simultaneously, acetic acid, propionic acid, and butyric acid levels rose, helping to inhibit inflammation. In summary, polysaccharides could regulate the host’s intestinal microbiota and inhibit inflammation.

### 3.6. As a Drug Carrier

The current first-line drugs for UC treatment in clinics often remain significant challenges due to their nontargeting therapeutic efficacy and severe side effects. Therefore, a straightforward disease-targeted drug delivery system is desirable. Based on the pathophysiological changes associated with UC, novel site-specific targeted drug delivery strategies that deliver drugs directly to the inflammation sites can enhance drug accumulation and decrease side effects. Nowadays, diverse encapsulated systems, including nanoparticles, inorganic particles, liposomes, prodrugs, hybrid systems, enteric-coated microneedle pills, enemas, and biological delivery systems, have been developed for drug delivery system in treating intestinal diseases [82]. Recent studies showed that the most widely used drug delivery platform was polymeric nanoparticles (NPs) [83]. Nanocarriers, including liposomes, nanoparticles, and micelles, have been widely used to prevent and treat UC in recent years [82]. NPs tend to be phagocytosed by macrophages and neutrophils at sites of inflammation and during interactions with the mucosal layer, which is thinner in inflamed tissues [84]. Using nanocarriers to encapsulate the first-line drugs for UC treatment could significantly reduce the toxicity [85], and it also could improve the therapeutic effect of drugs by taking advantage of the gap between intestinal cells resulting from the compromised integrity of the colonic epithelium in UC patients [86]. With obvious safety and biocompatibility, natural polysaccharides have been widely applied in nanocarriers as external materials. 

Natural polysaccharides have abundant chemical groups, such as hydroxyl, carboxyl, and amino groups, which can be chemically modified to improve their hydrophobicity. Xu et al. [86] fabricated colon-targeting nanoparticles based on *Angelica sinensis* polysaccharide specifically to release the naturally active compound ginsenoside Rh2 in the colonic inflammatory site (Figure 3A left), which significantly alleviated UC symptoms and improved gut microbial homeostasis (Figure 3A right). Moreover, the co-assembly of *Berberine-rhubarb* polysaccharide-NPs (BD) (Figure 3B left) was demonstrated to ameliorate the symptoms of the UC mouse model effectively, which is induced by DSS through regulating gut microbiota and amending the gut barrier integrity (Figure 3B right). This is due to the fact that BD has a longer retention time on the colon tissue and reacts with the microbiota and mucus in a thorough manner [87]. In addition, *Phragmites rhizoma* polysaccharide-based nanocarriers could protect azathioprine liposomes against gastrointestinal digestion, enhance the therapeutic effects on ulcerative colitis, and significantly reduce liver damage from azathioprine, which improve the efficacy and toxicity of drugs (Figure 3C) [88]. Zhu et al. [89] prepared Se nanoparticles (SeNPs) coated with *Ulva lactuca* polysaccharide (ULP-SeNPs) to treat UC. ULP-SeNPs exhibited anti-inflammatory effects to reduce the symptoms of acute colitis by inhibiting the hyperactivation of NF-κB in colonic tissues and macrophages. 

The polysaccharide-based nanocarrier can assist the drug in resisting the harsh environment of the gastrointestinal tract, improving stability, and concentrating on the regions of intestinal inflammation as much as possible. This effectively reduces drug side effects and enhances the drug’s bioavailability. Specific polysaccharides, acting as prebiotics, can confer upon drug nano-delivery systems the capacity to target the colon, based on enzyme-responsive properties. Furthermore, they can collaborate with drugs to mitigate IBD, due to their favorable anti-inflammatory activity and intestinal microecological regulation [86]. In conclusion, nanoparticle formulation, coated with polysaccharides from natural plants, may be a candidate for further developed drugs as potential therapeutic products for the treatment of UC. Nanoparticles coated with polysaccharides are likely to produce promising results in the treatment of UC. However, the applicability has not yet been well established. Hence, it is necessary to investigate in depth. 

## 4. Discussion

Polysaccharides from herbal plants are effective in the treatment of UC. There are numerous polysaccharides mentioned in this article. This paper summarizes that polysaccharides from herbal plants can act on different pathways to alleviate UC. Polysaccharides from *Schisandra chinensis* and Aloe could regulate the balance of inflammatory cytokines; polysaccharides *Pseudostellariae*, *Atractylodes macrocephala*, *Poria cocos* could regulate the balance of immune cells; polysaccharides from Noni fruit, *Scutellaria baicalensis*, *Gracilaria lemaneiformis* could regulate the function of intestinal mucus; polysaccharides from *Morinda citrifolia*, *Dendrobium officinale* could protect the intestinal barrier integrity; polysaccharides from Aloe, *Lonicera japonica*, *Chrysanthemum morifolium* could regulate the intestinal microbiota. In addition, nanoparticles could be coated with different polysaccharides and used to treat UC as was shown with polysaccharides from *Angelica sinensis* and *Ulva lactuca*. In conclusion, these findings discussed in the review provide a theoretical basis for applying natural plant polysaccharides in the treatment of UC.

Polysaccharides from natural herbal plants have several advantages for UC treatment, including safety, definite effects, and few adverse reactions. Polysaccharides can alleviate UC in various ways. In addition, they can be combined with other drugs for comprehensive treatment. In conclusion, these findings provide a solid scientific basis for utilization of natural polysaccharides from herbal plants as promising candidates for UC therapy. Nevertheless, the therapeutic effects of the polysaccharides discussed in the article have only been demonstrated in animal models. More in vivo studies, especially in humans, are warranted for further elucidation. Consequently, further clinical and preclinical studies are required to determine the molecular mechanisms, thus facilitating the clinical translation of natural polysaccharides for the treatment of UC; only so that polysaccharides could be further used in the treatment of UC. 

However, some challenges and problems remain unsolved. As macromolecules, polysaccharides have a complex structure. Moreover, many biological activities of polysaccharides are closely related to their structure. There is an increasing number of studies on the biological activity of polysaccharides, but few studies on the relationship between their structure and immune-regulatory activity [90]. It is necessary to investigate the different structures and bioactivities of these polysaccharides to identify drugs with optimal therapeutic efficacy, targeting, and bioavailability. The evaluation criteria for the efficacy and safety of polysaccharides in the treatment of UC are not yet standardized, and the research on their therapeutic mechanisms is not sufficiently in depth. To improve the therapeutic efficacy of UC and promote the development of new drugs, research needs to be standardized and rationalized. In addition, polysaccharides can be combined with other drugs for comprehensive treatment, and targeted formulations are expected to facilitate the development of polysaccharide-based drugs with targeted delivery and high efficacy, which has been a hot research topic in recent years.

## Figures and Tables

**Figure 1 pharmaceutics-16-01073-f001:**
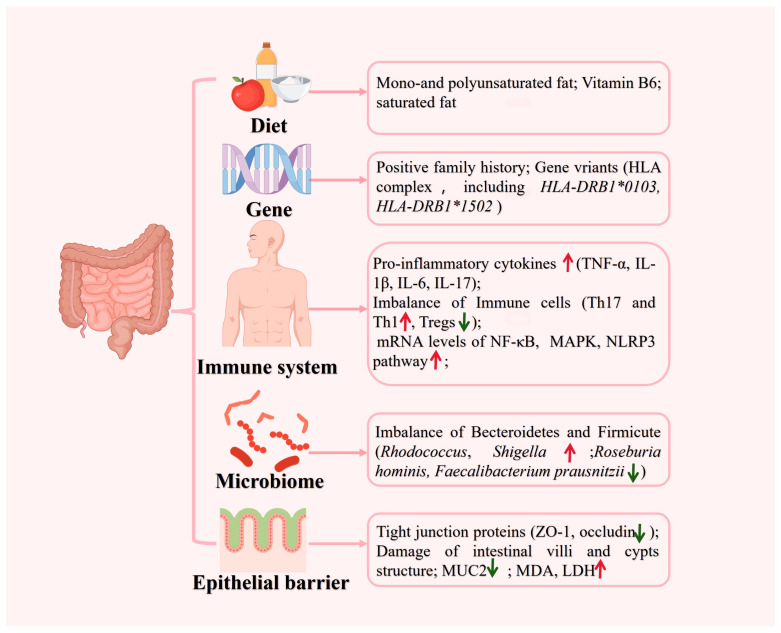
Main risk factors of UC. The image was prepared by Figdraw 2.0.

**Figure 2 pharmaceutics-16-01073-f002:**
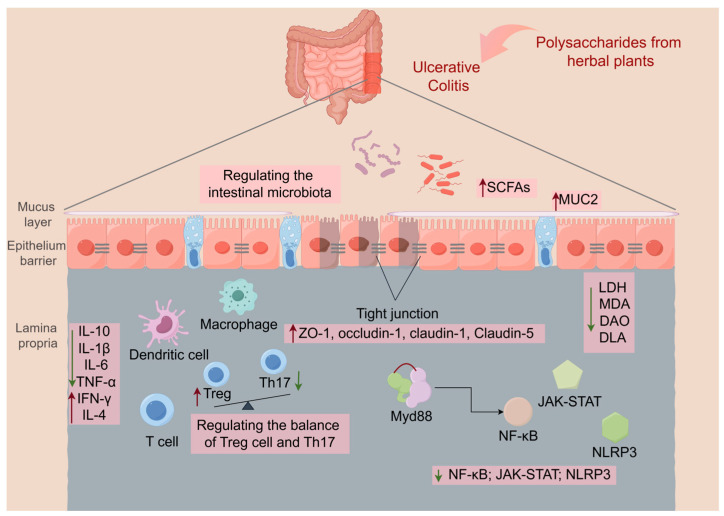
The mechanism of polysaccharides from Chinese medicine herbal plants in UC. The image was prepared by Figdraw 2.0.

**Figure 3 pharmaceutics-16-01073-f003:**
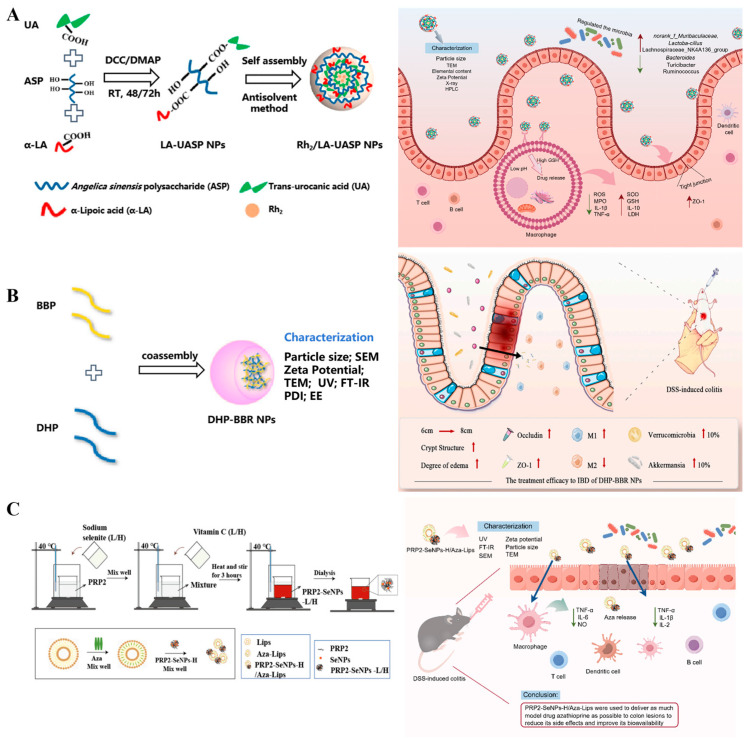
The preparation process of polysaccharide-based nanoparticle and its mechanism in UC. (**A**) The preparation process of Rh2/LA-UASP (left) and its bioactivity in UC (right) [86]. (**B**) The preparation process of BD (left) and its treatment efficacy in colon(right) [87]. (**C**) The preparation process of PRP2-SeNPs-L/H and PRP2-SeNPs-H/Aza-Lips (left) and its bioactivity in UC (right) [88]. The image was prepared by Figdraw 2.0.

**Table 1 pharmaceutics-16-01073-t001:** The immune regulation effects of polysaccharides from different herbal plants in UC.

Source	PS	The Role in UC Treatment	Refs.
Chrysanthemum	CP	Decreased inflammatory cytokines Decreased mRNA levels of TLR4, NF-κB, IL-6, STAT3, and JAK2 Decreased The expression of p-P65, TLR4, p-STAT3, and p-JAK2	[42]
*Scutellaria baicalensis*	SP1-1	Inhibited expression of TNF-α, IL-1β, IL-6, IL-17 Decreased CD11b^+^ macrophage infiltration in colons Decreased Cle-caspase-1 Decreased *Bacteroides*, *Proteobacteria* and *Staphylococcus*	[43]
*Astragalus*	APS	Inhibited NF-κB phosphorylation Decreased e and MPO	[44]
*Tremella fuciformis*	TFP	Decreased inflammatory cells infiltration Restored intestinal epithelial barrier integrity Decreased TNF-α, IL-1β and IL-6 Improved mRNA and protein expression of ZO-1 and OCLN	[45]
*Dendrobium officinale*	DOPS	Improved clinical signs and symptoms Decreased mortality Alleviated colonic pathological damage Reduced the level of IL-1β, IL-6, IL-18, TNF-α and IFN-γ	[46]
Aloe	AP	Down-regulated IL-6, JAK2, STAT-3 and cell apoptosis	[47]
*Lycium barbarum*, *Astragalus membranaceus*	QHPS	Inhibited levels of DAO, DLA and EDT Protected the integrity of the intestinal mucosa Repaired the injury of intestinal tract	[48]
*Schisandra chinensis*	SCP	Decreased IL-6, IL-10, IL-17, IL-23, and TNF-α levels Returned abundance of *Firmicutes*, *Proteobacteria*, and *Bacteroidetes* Increased the content of SCFAs	[49]
Noni fruit	NFP	Promoted the expression of zonula, occludens-1	[50]
*Arctium lappa*	ALP-1	Decreased IL-1β, IL-6 and TNF-α and IL-10 Increased the abundance of *Firmicutes*, *Ruminococcaceae*, *Lachnospiraceae*, *Lactobacillus* Inhibited the level of *Proteobacteria*, *Alcaligenaceae*, *Staphylococcus and* and *Bacteroidetes*	[51]
*Scutellaria baicalensis*	SP2-1	Suppressed proinflammatory cytokines. Up-regulated expressions of ZO-1, OCLN and Claudin-5; Enhanced the levels of acetic acid, propionic acid, and butyric acid Increased the abundance of *Firmicutes*, *Bifidobacterium*, *Lactobacillus*, and *Roseburia* Inhibited the levels of *Bacteroides*, *Proteobacteria* and *Staphylococcus*	[43]
*Rheum tanguticum*	RTP	Elevated cell survival, SOD activity Decreased MDA, LDH activity and cell apoptosis	[52]
Glucomannans	GMs	Reduced IL-1β, IL-6 and TNF-α and IL-10 Regulated the expressions of TLR-2, TLR-4, TLR-6, and TLR-9	[53]
*Chrysanthemum morifolium*	-	Decreased the abundances of *Enterococcus*, *Escherichia* and *prevotella* Eelevated *Bifidobacterium*, *Butyricicoccus*, *Clostridium*, *Lachnospiraceae*, *Lactobacillus* and *Rikenellaceae*	[54]
*Gracilaria Lemaneiformis*	SP	Suppressed the secretion of TNF-α, IL-6 and IL-1β Promoted Claudin-1, ZO-1, and MUC-2	[55]
Tamarind xyloglucan	TCG	Decreased IL-1β and IL-6 levels Decreased the expression of TLR4, MyD88, I-κB and NF-κB	[56]
*Angelica sinensis*	ASP	Reduced IL-1β, IL-6 levels Improved occludens 1, occludin, and claudin-1	[57]
*Pseudostellaria heterophylla*	PAMK	Decreased IL-1β and TNF-α Increased the abundance of *Bacteroides*	[58]
*Atractylodes macrocephala*	AMP	Regulated the balance between Th 17 and the Treg cells; Increased the content of SCFAs; Increased *Butyricicoccus, Lactobacillus*, Decreased *Actinobacteria*, *Akkermansia*, *Anaeroplasma*, *Bifidobacterium*, *Erysipelatoclostridium*, *Faecalibaculum*, *Parasutterella*, *Parvibacter*, *Tenericutes*, *Verrucomicrobia*	[59,60]

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
