# Peer review of "The Potential Role of Plant Polysaccharides in Treatment of Ulcerative Colitis"

_pharmaceutics, 2024, doi:10.3390/pharmaceutics16081073_

Round 1

Reviewer 1 Report

Comments and Suggestions for Authors

I felt that the authors report contained a complete and accurate description of the design methods and conclusion and I no further comments. I do feel that the authors should include a discussion o the ketamine and xyclazine anesthetic influence on the result and add a control study using ether or pentobarbital.

Suggest inclusion of in vivo colitis models

Author Response

Comments1. I felt that the authors report contained a complete and accurate description of the design methods and conclusion and I no further comments. I do feel that the authors should include a discussion o the ketamine and xyclazine anesthetic influence on the result and add a control study using ether or pentobarbital

Response1. We are very grateful for the reviewer’s comments. Anaesthetics such as ketamine and thiazide have not been used to establish a mouse model of ulcerative colitis. Cyclophosphamide, acetic acid, are commonly used to create models. Most of the literature cited in the article used cyclophosphamide and no anaesthetics were used in the experiments.

Reviewer 2 Report

Comments and Suggestions for Authors

Review of The Potential Role of Plant Polysaccharides in Treatment of Ulcerative Colitis 

Abstract  Abstract is missing in reviewers’ pdf copy. 

Introduction

Line 24-25 – Do authors mean to say “treatment” instead of “cure”? To my knowledge, there is no cure to UC currently, but there may be a “dramatic increase” in types of treatments.

Line 27 – Almost appears like a factual claim? Could authors provide a reference to this statement?

Line 29 – Authors include “biological agents” in a sentence that contains examples of them. Could you please consider amending your sentence to “biological agents, such as”.

Additionally, “microecologics” is grammatically incorrect, despite occurring in ISI journals on occasion. In the ISI database, Web of Science, this strange word only occurs in some 14 reports, and all of them issued from China. Please consider amending this term to micro-ecologics, or micro ecologics.

 Line 31-32 – Authors should include in text that … herbal medicines, while safe, also have varied and often limited efficacy, like the drugs listed above … to avoid a biased statement.

Line 37 – Remove “as we all know”. There is no room in scientific English, while reporting observations of actual experiments, for such nuanced and dramatic terms.

 Line 40 – “Based on its absorbed in the intestine”, unclear meaning. Please revise the manuscript with more precise care.

Line 44-46 – Sentence meaning unclear. Consider revising to achieve more clarity in understanding for the reader.

 Section 2

Line 61-62 – Incorrect tense use, sentence error.

Figure 1 – There are many typos in the figure text. Could authors remove the faded cell images in the background as they are not adding to the interpretation of the figure, and as they confuse the scale. Please remove “etc” in the figure text. This Latin abbreviation is not a standard term in Scientific English writing. Also, it would be helpful if you consider to include a brief description of the main figure points in the figure legend.

Line 67 – “Are shift”, incorrect grammar.

Line 80-81 – “in vivo experimental and colitis models”, “and” is redundant.

Line 87 – remove capital S from studies.

 Section 3

Line 117 – change “has” to “have”.

Line 123 – change “accumulating” to “accumulation”.

Line 127 – instead of “discussed” do authors mean “summarized”?

Figure 2 – similar to Figure 1, the background of the figure distracts from your main message.

Line 133 – “As a kind of low…” should be modified. There is no room in scientific reports for colloquial phrases or jargon.

Lines 218-219 – the opening statement appears disconnected from the discussion in the rest of paragraph. What are we reading here? Make this paragraph more clear and legible.

Section 3.6, first paragraph – authors summarize that nanocarriers are widely used and indicate safe, efficacious use in UC. I would suggest addition of more hypothetical interpretation as many of the patents and delivery systems proposed and summarized (e.g. those from reference “92”), include UC-treatment strategies in early clinical trials and new technologies. So, while promising, the applicability is not well established yet.

Lines 322-324 – acceptable conclusion.

Figure 3, line 327 – do authors mean “its bioactivity in UC”? The panels on the right side of the figure are hard to read due to low resolution.

Overall, the authors have provided a nice summary of the existing experimental work on plant polysaccharides and their potential impact on UC. While this manuscript in its present form attempts to provide an overview, the lack of in-depth analysis of the reviewed experiments’ caveats or limitations means that the overall results presented may not be an accurate representation of the true effect in vivo. The authors should expand on this limitation in their final discussion to indicate that the studies reviewed here may have limitations that should be considered in results, but the potential outcome is promising.

Comments on the Quality of English Language

Check that all your science-related terms are the standard reference terms, by checking for their rate of occurrence in Clarivate Analytics. Try not to use newly-invented words like microecologics as their meaning is not decipherable.

Revise the prose to be correct grammatically, avoiding non-scientific English. As we all know, there are over 200 countries around the world speaking thousands of different languages, but Scientific English is the reference standard. So please try to revise the manuscript to achieve perfect undebatable clarity of thought.

Author Response

Comments 1. Abstract is missing in reviewers’ pdf copy.

Response 1. We apologize for missing information. We hope to post the full version this time.

Comments 2. Do authors mean to say “treatment” instead of “cure”? To my knowledge, there is no cure to UC currently, but there may be a “dramatic increase” in types of treatments.

Response2.We apologize for the mistake and have revised the “cure” to “treatment”.

We are very grateful for the reviewer’s comments. We agree with the reviewer's suggestion. Polysaccharides may provide some alleviation of ulcerative colitis (UC) symptoms through anti-inflammatory effects or by regulating the microbiome of the gut. However, this is unlikely to be a cure. We have revised the “cure” to “treatment”, please check it. Line 23-24” In the past decade, the treatment for UC has increased dramatically with the advent of biologics and small molecules, and clinical and endoscopic remission of these treatments are also employed.”

Comments 3. Almost appears like a factual claim? Could authors provide a reference to this statement?

Response3. We have added a reference for this sentence. It marked in red, please check it. Line 26-27 and reference1.

Comments 4. Authors include “biological agents” in a sentence that contains examples of them. Could you please consider amending your sentence to “biological agents, such as”.Additionally, “microecologics” is grammatically incorrect, despite occurring in ISI journals on occasion. In the ISI database, Web of Science, this strange word only occurs in some 14 reports, and all of them issued from China. Please consider amending this term to micro-ecologics, or micro ecologics.

Response 4. We appreciate the reviewer’s comments. Example of Biological agents have been given in the article. (Line 28-29) It is marked in red. And we have rivesed the “microecologics” to “micro-ecologics”.

Line 27-28 “At present, the drugs for UC mainly include 5-amino salicylic acids, glucocorticoids, immunosuppressants, micro-ecologics, biological agents, such as tumor necrosis factor (TNF) antagonists, anti-integrin agents and Janus kinase (JAK) inhibitors. ”

Comments 5. Line 31-32 – Authors should include in text that … herbal medicines, while safe, also have varied and often limited efficacy, like the drugs listed above … to avoid a biased statement.

Response 5. Thanks to the reviewers' suggestion. I have mentioned in discussion part of text. Check it in line 352-353“However, some challenges and troubles have been unsolved. Polysaccharides also have varied and often limited efficacy. ”

Comments 6. Line 37 – Remove “as we all know”. There is no room in scientific English, while reporting observations of actual experiments, for such nuanced and dramatic terms.

Response 6. We apologize for the mistake and have revised it. Please check it line 38.

Comments 7. Line 40 – “Based on its absorbed in the intestine”, unclear meaning. Please revise the manuscript with more precise care.

Response 7. We apologize for the mistake and have revised “based on its absorbed in the intestine” to “Based on whether it’s absorbed in intestine”. Line 40, It is marked in red. Please check it.

Comments 8. Line 44-46 – Sentence meaning unclear. Consider revising to achieve more clarity in understanding for the reader.

Response 8. We appreciate the reviewer’s comments. We have revised these sentences and making it more understanding. It is marked in red. Please check it in line 43-46

“A proportion of polysaccharides are also resistant to digestion and are mainly metabolized by the gut microbiota in the large intestine, modulating the gut microbiota diversity and compositional structure of the gut microbiota.”

Comments 9. Line 61-62 – Incorrect tense use, sentence error.

Response 9. We apologize for the mistake and have revised it. It is marked in red. Please check it.

Line 60-61 “A positive family history has been reported between 1.5 and 24% in UC”

Comments 10. Figure 1 – There are many typos in the figure text. Could authors remove the faded cell images in the background as they are not adding to the interpretation of the figure, and as they confuse the scale. Please remove “etc” in the figure text. This Latin abbreviation is not a standard term in Scientific English writing. Also, it would be helpful if you consider to include a brief description of the main figure points in the figure legend.

Response 10. Thanks to the reviewers' suggestion. For the Figure 1, we have removed the faded cell image. We have mentioned a brief description of Figure 1 in “Etiology and pathogenesis of UC” part, hence we didn’t mention in the figure.

Comments 11. Line 67 – “Are shift”, incorrect grammar.

Response 11. We apologize for the mistake and have revised it. It is marked in red. Please check it.

Line 67 The main features, gut microbiota (GM) and gut-associated lymphoid tissue (GALT) are shifting towards an inflammatory pattern, for example T regulatory (Tregs) cells in GALT are reduced, with an increase in T helper cell 17 (Th17) and T helper 1 (Th1) cells and enhancement of tumor necrosis factor (TNF) and interleukin-1β (IL-1β) production.

Comments 12. Line 80-81 – “in vivo experimental and colitis models”, “and” is redundant.

Response 12. We apologize for the mistake and have revised it. It is marked in red. Please check it.

Line 80-81 “The increasing number of studies on colitis models have identified the important role of TJs proteins in the pathogenesis of UC”

Comments 13. Line 87 – remove capital S from studies.

Response 13. We apologize for the mistake and have revised it.

Comments 14. Line 117 – change “has” to “have”.  Line 123 – change “accumulating” to “accumulation”.

Response 14. We apologize for the mistake and have revised the “has” to “have”, “accumulating” to “accumulation”.

Line 117. In recent years, natural polysaccharides for protecting intestinal health have become a topic of intestine research.

Line 122. Zhang Y et al. found that polysaccharides could be absorbed through the small intestine followed by accumulation in the liver and kidney.

Comments 15. Line 127 – instead of “discussed” do authors mean “summarized”? Figure 2 – similar to Figure 1, the background of the figure distracts from your main message.

Response 15. We apologize for the mistake and have revised the “discussed” to “summarized”, For the figure 2, we have removed the faded cell images.

Line 126. The detailed mechanisms of plant-derived polysaccharides in UC were summarized as follows.

Comments 16. Lines 218-219 – the opening statement appears disconnected from the discussion in the rest of paragraph. What are we reading here? Make this paragraph more clear and legible.

Response 16. We sincerely thank the reviewers for their valuable comments, and we have checked this paragraph. We have changed the paragraph name to “Protecting the intestinal barrier integrity”. Intestinal barrier formed by intestinal epithelial cells (IECs). The maintenance of the intestinal epithelial barrier is the essential function of the IECs[1]. Therefore, in this paraph we mentioned the malondialdehyde (MDA), lactate dehydrogenase (LDH), which indicate the apoptosis of epithelial apoptosis. Polysaccharides could reduce the apoptosis of epithelial cells, thereby protect the intestinal barrier integrity.

Line 224-226 “In the gastrointestinal tract, apoptosis is typically limited to superficial epithelial cells. However, apoptosis can be more extensive in pathological states of inflammation or infection”

Comments 17. Section 3.6, first paragraph – authors summarize that nanocarriers are widely used and indicate safe, efficacious use in UC. I would suggest addition of more hypothetical interpretation as many of the patents and delivery systems proposed and summarized (e.g. those from reference “92”), include UC-treatment strategies in early clinical trials and new technologies. So, while promising, the applicability is not well established yet.

Response 17. Thank you for your thorough review. In response to the reviewer's suggestions, we added relevant content.

Line 330-332“Nanoparticles coated with polysaccharides are likely to produce promising results in the treatment of UC. However the applicability is not well established yet. Hence it is necessary to investigate in-depth.”

Comments 18. Figure 3, line 327 – do authors mean “its bioactivity in UC”? The panels on the right side of the figure are hard to read due to low resolution.

Response 18. Thanks to the reviewers' suggestion. Its means that “its bioactivity in UC”. We have improved resolution of the image. Please check it.

Comments 19. Overall, the authors have provided a nice summary of the existing experimental work on plant polysaccharides and their potential impact on UC. While this manuscript in its present form attempts to provide an overview, the lack of in-depth analysis of the reviewed experiments’ caveats or limitations means that the overall results presented may not be an accurate representation of the true effect in vivo. The authors should expand on this limitation in their final discussion to indicate that the studies reviewed here may have limitations that should be considered in results, but the potential outcome is promising.

Response 19. We are very grateful for the reviewer’s comments. We have added relevant details about the limitations in discussion part. It marked in red. Please check it.

Line 339-345 “In conclusion, these findings provided a solid scientific basis for utilization of natural polysaccharides from herbal plants, as promising candidates for UC therapy. Nevertheless, the therapeutic effects of the polysaccharides discussed in the article have been demonstrated in animal models. Consequently, further clinical and preclinical studies are required to elucidate the molecular mechanisms, thus facilitating the clinical translation of natural polysaccharides for the treatment of UC.”

Line 351-356 “It is necessary to investigate the different structure and bioactivities of these polysaccharides in order to identify drugs with optimal therapeutic efficacy, targeting and bioavailability. In addition, polysaccharides can be combined with other drugs for comprehensive treatment and targeted formulations is expected to facilitate the development of polysaccharides-based drugs with targeted delivery and high efficacy, which has been a hot research topic in recent years. ”

Comments 20. Comments on the Quality of English Language

Check that all your science-related terms are the standard reference terms, by checking for their rate of occurrence in Clarivate Analytics. Try not to use newly-invented words like microecologics as their meaning is not decipherable.

Revise the prose to be correct grammatically, avoiding non-scientific English. As we all know, there are over 200 countries around the world speaking thousands of different languages, but Scientific English is the reference standard. So please try to revise the manuscript to achieve perfect undebatable clarity of thought.

Response 20. Thanks to the reviewers' suggestions, we have done our best to improve the manuscript and have carefully and thoroughly checked it for grammatical errors, and we did invite a native English-speaking professional to touch up the manuscript, and these changes will not affect the content or framework of the paper.

Reviewer 3 Report

Comments and Suggestions for Authors

The manuscript submitted by Dilixiati and colleagues summarizes the available and recent knowledge on the role of different plant polysaccharides for the treatment of ulcerative colitis. The review article is well illustrated and properly structured. It will attract the interest of researchers working in the field. Important part of the manuscript is the subsection “As a drug carrier” that describes recent advances in the development of drug-delivery systems based on polysaccharide nanoparticles or polysaccharide containing nanoparticles and their potential role for the treatment of UC. However, a major drawback of this review is the lack of conclusions and the short discussion. In addition, there are several recent review articles on the same topic (10.1016/j.carbpol.2020.117189; 10.3390/molecules27196467; 10.3389/fphar.2022.927855). Therefore, the authors should explain in the abstract and in the discussion the benefits and advance of their article. The authors should pay attention to the following main points:

1)      It is necessary to provide a hypothesis on the plant polysaccharide potential for the treatment of UC and highlight perspective for future improvement of UC therapy. A review article “should be critical and constructive and provide recommendations for future research”.

2)     Are there reports describing the relationship between polysaccharide structure and their mechanism of action? Please discuss more thoroughly with regard to UC potential treatment.

3)     Are there any synergistic effects reported if plant polysaccharides and drugs or drug-loaded polymeric nanoparticles are applied?

4)     There are numerous stylistic errors in the text. Please thoroughly revise the manuscript according to the comments in the attached file.

Minor points:

1)     Abbreviations should be defined the first time they appear in each of the three sections: the abstract; the main text; the first figure or table. A list with all abbreviations used in the text will be very useful for the reader. There are many abbreviations and some of them have not been defined (DSS, CTX per example)

2)     Duplicated sentence: L8-9, L20-22. Delete the sentence in the abstract.

Comments on the Quality of English Language

Comments on the quality of English are shown in the attached file.

Author Response

Comments 1. It is necessary to provide a hypothesis on the plant polysaccharide potential for the treatment of UC and highlight perspective for future improvement of UC therapy. A review article “should be critical and constructive and provide recommendations for future research”.

Response 1. We are very grateful for the reviewer’s comments. We have added relevant details.

Line 339-345“In conclusion, these findings provided a solid scientific basis for utilization of natural polysaccharides from herbal plants, as promising candidates for UC therapy. Nevertheless, the therapeutic effects of the polysaccharides discussed in the article have been demonstrated in animal models. Consequently, further clinical and preclinical studies are required to elucidate the molecular mechanisms, thus facilitating the clinical translation of natural polysaccharides for the treatment of UC.”

Line 351-356

“It is necessary to investigate the different structure and bioactivities of these polysaccharides in order to identify drugs with optimal therapeutic efficacy, targeting and bioavailability. In addition, polysaccharides can be combined with other drugs for comprehensive treatment and targeted formulations is expected to facilitate the development of polysaccharides-based drugs with targeted delivery and high efficacy, which has been a hot research topic in recent years.”

Comments 2. Are there reports describing the relationship between polysaccharide structure and their mechanism of action? Please discuss more thoroughly with regard to UC potential treatment.

Response2. Thanks to the reviewers' suggestions. There is an increasing number of studies on the biological activity of polysaccharides, but few studies on the relationship between its structure and immune-regulatory activity. Hence, this is a limitation for use of polysaccharides. We added relevant content in text.

Line 351-360 “However, there are some challenges and troubles that are unsolved. As macromolecules, polysaccharides have a complex structure. Moreover, many biological activities of polysaccharides are closely related to their structure. There is an increasing number of studies on the biological activity of polysaccharides, but few studies on the relationship between its structure and immune-regulatory activity. It is necessary to investigate the different structure and bioactivities of these polysaccharides to identify drugs with optimal therapeutic efficacy, targeting and bioavailability.”

Comments 3. Are there any synergistic effects reported if plant polysaccharides and drugs or drug-loaded polymeric nanoparticles are applied?

Response 3. Thank you for your thorough review. The are some reports on synergistic effects. For example, this article[1] constructed Phragmites rhizoma polysaccharide-based nano-drug delivery systems (PRP2-SeNPs-H/Aza-Lips) for synergistically alleviating ulcerative colitis and to investigate the important roles of Phragmites rhizoma polysaccharide-based nanocarriers in PRP2-SeNPs-H/Aza-Lips. In another review have discussed that Polysaccharide-based nanocarriers with high safety and bioavailability are often used in the construction of colon-targeted drug nano delivery systems (DNSs)[2]. It can help the drug resist the harsh environment of gastrointestinal tract, improve stability and concentrate on the intestinal inflammation regions as much as possible, which effectively reduces drug side effects and enhances its bioavailability. Certain polysaccharides, as prebiotics, can not only endow DNSs with the ability to target the colon based on enzyme responsive properties, but also cooperate with drugs to alleviate IBD due to its good anti-inflammatory activity and intestinal microecological regulation.

Comments 4. There are numerous stylistic errors in the text. Please thoroughly revise the manuscript according to the comments in the attached file.

Response 4. We apologize for the mistake. We have revised the manuscript according to the attached file. We have done our best to improve the manuscript and checked it for stylistic error.

Comments 5. Abbreviations should be defined the first time they appear in each of the three sections: the abstract; the main text; the first figure or table. A list with all abbreviations used in the text will be very useful for the reader. There are many abbreviations and some of them have not been defined (DSS, CTX per example)

Response 5. Thanks to the reviewers' suggestions. We apologize for the mistake. We have defined the abbreviations in the text. It is marked in red, please check it.

Comments 6. Duplicated sentence: L8-9, L20-22. Delete the sentence in the abstract.

Response 6. We apologize for the mistake. We have delated the sentence in the abstract.          Please check it.

References:

  1. Feng, Y.; Wu, C.; Chen, H.; Zheng, T.; Ye, H.; Wang, J.; Zhang, Y.; Gao, J.; Li, Y.; Dong, Z. Rhubarb Polysaccharide and Berberine Co-Assembled Nanoparticles Ameliorate Ulcerative Colitis by Regulating the Intestinal Flora. Front Pharmacol 2023, 14, 1184183, doi:10.3389/fphar.2023.1184183.
  2. Cui, M.; Zhang, M.; Liu, K. Colon-Targeted Drug Delivery of Polysaccharide-Based Nanocarriers for Synergistic Treatment of Inflammatory Bowel Disease: A Review. Carbohydrate Polymers 2021, 272, 118530, doi:10.1016/j.carbpol.2021.118530.

Round 2

Reviewer 3 Report

Comments and Suggestions for Authors

The revised manuscript by Dilixiati and colleagues shows a slight improvement but most of my comments and recommendations have not been considered. To facilitate the authors, I have uploaded a file with comments, marked words and phrases that need correction. Regrettably, few of my remarks have been taken into account by the authors. I have expressed my opinion and highlighted the major drawbacks of the manuscript. I will repeat these points because I don’t see sufficient revision regarding them:

1)     A major drawback of the review is the lack of conclusions and the short discussion. The discussion has been revised but needs further improvement. The conclusions are still lacking.

2)     There are several recent review articles on the same topic (10.1016/j.carbpol.2020.117189; 10.3390/molecules27196467; 10.3389/fphar.2022.927855). Therefore, the authors should explain in the abstract and in the discussion the benefits and advance of their article. It is necessary to provide a hypothesis on the plant polysaccharide potential for the treatment of UC and highlight perspective for future improvement of UC therapy. A review article “should be critical and constructive and provide recommendations for future research”.

3)     Please discuss more thoroughly the synergistic effects reported so far regarding application of plant polysaccharides together with drugs or application of plant polysaccharides together drug-loaded polymeric nanoparticles.

Comments on the Quality of English Language

My comments have been indicated in the attached file during the first review round.

Author Response

Comments1:A major drawback of the review is the lack of conclusions and the short discussion. The discussion has been revised but needs further improvement. The conclusions are still lacking.

Response1: We are very grateful for the reviewer’s comments. In response to the reviewer's suggestions, we added relevant content, it is marked in yellow.

Line 32-34: Therefore, there is an urgent need to discover new therapies that are both curable and tolerable for patients with UC. Based on the shortcomings of conventional drugs, research and development of herbal medicine have been initiated in the field of UC treatment.

Line 159-161: In conclusion, polysaccharides extracted from plants could regulate the balance of proinflammatory and anti-inflammatory cytokines, which is an important way of alleviating the UC.

Line 350-360: Polysaccharides from herbal plant are effective in the treatment of UC. There are numerous polysaccharides that are mentioned in the article. Polysaccharides from Schisandra chinensis, Aloe could regulate the balance of inflammatory cytokines; polysaccharides Pseudostellariae, Atractylodes macrocephala, Poria cocos could regulate the balance of immune cells; polysaccharides from Noni fruit, Scutellaria baicalensis, Gracilaria lemaneiformis could regulate the function of intestinal mucus; polysaccharides from Morinda citrifolia, Dendrobium officinale could protect the intestinal barrier integrity, polysaccharides from Aloe, Lonicera japonica, Chrysanthemum morifolium could regulate the intestinal microbiota. In addition, nanoparticles could coat with polysaccharides from Angelica sinensis, Ulva lactuca to treat UC.

Comments2:There are several recent review articles on the same topic (10.1016/j.carbpol.2020.117189; 10.3390/molecules27196467; 10.3389/fphar.2022.927855). Therefore, the authors should explain in the abstract and in the discussion the benefits and advance of their article. It is necessary to provide a hypothesis on the plant polysaccharide potential for the treatment of UC and highlight perspective for future improvement of UC therapy. A review article “should be critical and constructive and provide recommendations for future research”.

Response: Thanks to the reviewers' suggestion. I have mentioned in discussion part of text.

Line 364-371: In conclusion, these findings provided a solid scientific basis for utilization of natural polysaccharides from herbal plants as promising candidates for UC therapy. Nevertheless, the therapeutic effects of the polysaccharides discussed in the article have been demonstrated in animal models. More in vivo studies, especially in humans, are warranted to further elucidate. Consequently, further clinical and preclinical studies are required to elucidate the molecular mechanisms, thus facilitating the clinical translation of natural polysaccharides for the treatment of UC. Only so that polysaccharides could be further used in the treatment of UC.

Line 378-382, The evaluation criteria for the efficacy and safety of polysaccharides in the treatment of UC are not yet standardized, and the research on their therapeutic mechanisms is not sufficiently in-depth. To improve the therapeutic efficacy of UC and promote the development of new drugs, research needs to be standardized and rationalized.

Comments3:Please discuss more thoroughly the synergistic effects reported so far regarding application of plant polysaccharides together with drugs or application of plant polysaccharides together drug-loaded polymeric nanoparticles.

Response: Thanks to the reviewers' thoughtful suggestions, we added relevant content, it is marked in yellow.

Line331-338: The polysaccharide-based nanocarrier can assist the drug in resisting the harsh environment of the gastrointestinal tract, improving stability and concentrating on the regions of intestinal inflammation as much as possible. This effectively reduces drug side effects and enhances the drug's bioavailability. Specific polysaccharides, acting as prebiotics, can confer upon drug nano-delivery systems the capacity to target the colon based on enzyme-responsive properties. Furthermore, they can collaborate with drugs to mitigate IBD, due to their favorable anti-inflammatory activity and intestinal microecological regulation.

Round 3

Reviewer 3 Report

Comments and Suggestions for Authors

The manuscript is improved and could be accepted for publication. However, in the new revised version there isn't a clear explanation in the abstract and in the discussion for the benefits and significance of this review article (Comment #2 in my previous report) . 

Minor points:

L10: "Polysaccharides" should be with lower case.

L349: Revise: Polysaccharides from herbal plants.

L352: Poria cocus should be italicized. L355: Dendrobium officinale should be italicized.

L357: Revise the sentence. I guess the authors mean that nanoparticles could be coated with different polysaccharides and used to treat UC as it has been shown with polysaccharides from Angelica sinensisUlva lactuca.

Comments on the Quality of English Language

The comments are listed in the previous section.

Author Response

Comments1 :However, in the new revised version there isn't a clear explanation in the abstract and in the discussion for the benefits and significance of this review article.

Response:. We are very grateful for the reviewer’s comments. In response to the reviewer's suggestions, we added relevant content, it is marked in green.

Line 15-16: This review provides a theoretical basis for applying natural plant polysaccharides in the prevention and treatment of UC.

Line 352-353: This paper summarizes that polysaccharides from herbal plants can act on different pathways to alleviate UC.

Line 360-361: In conclusion, these findings discussed in the review provides a theoretical basis for applying natural plant polysaccharides in the treatment of UC.

Line 365-367: In conclusion, these findings provided a solid scientific basis for utilization of natural polysaccharides from herbal plants as promising candidates for UC therapy.

Comments2:L349: Revise: Polysaccharides from herbal plants.

Response: We apologize for the mistake and have revised it. Please check it.

Comments3:L352: Poria cocus should be italicized. L355: Dendrobium officinale should be italicized.

Response: We apologize for the mistake and have revised it. Please check it.

Comments4:L357: Revise the sentence. I guess the authors mean that nanoparticles could be coated with different polysaccharides and used to treat UC as it has been shown with polysaccharides from Angelica sinensis, Ulva lactuca.

Response: We apologize for the mistake and have revised it. Please check it.
